# Identification of Potentially Therapeutic Target Genes of Hepatocellular Carcinoma

**DOI:** 10.3390/ijerph17031053

**Published:** 2020-02-07

**Authors:** Chengzhang Li, Jiucheng Xu

**Affiliations:** 1College of Life Science, Henan Normal University, Xinxiang 453007, China; lee125@126.com; 2Engineering Lab of Intelligence Business & Internet of Things, College of Computer and Information Engineering, Henan Normal University, Xinxiang 453007, China

**Keywords:** hepatocellular carcinoma, hub gene, PPI network, MCC algorithm, gene expression profile

## Abstract

Background: Hepatocellular carcinoma (HCC) is a major threat to public health. However, few effective therapeutic strategies exist. We aimed to identify potentially therapeutic target genes of HCC by analyzing three gene expression profiles. Methods: The gene expression profiles were analyzed with GEO2R, an interactive web tool for gene differential expression analysis, to identify common differentially expressed genes (DEGs). Functional enrichment analyses were then conducted followed by a protein-protein interaction (PPI) network construction with the common DEGs. The PPI network was employed to identify hub genes, and the expression level of the hub genes was validated via data mining the Oncomine database. Survival analysis was carried out to assess the prognosis of hub genes in HCC patients. Results: A total of 51 common up-regulated DEGs and 201 down-regulated DEGs were obtained after gene differential expression analysis of the profiles. Functional enrichment analyses indicated that these common DEGs are linked to a series of cancer events. We finally identified 10 hub genes, six of which (*OIP5*, *ASPM*, *NUSAP1*, *UBE2C*, *CCNA2*, and *KIF20A*) are reported as novel HCC hub genes. Data mining the Oncomine database validated that the hub genes have a significant high level of expression in HCC samples compared normal samples (*t*-test, *p* < 0.05). Survival analysis indicated that overexpression of the hub genes is associated with a significant reduction (*p* < 0.05) in survival time in HCC patients. Conclusions: We identified six novel HCC hub genes that might be therapeutic targets for the development of drugs for some HCC patients.

## 1. Introduction

Hepatocellular carcinoma (HCC), a major threat to public health, is the fifth most common cancer worldwide, causes about one million deaths each year [1,2]. HCC mortality rates have risen over the last few decades in most countries [3]. HCC can be managed by the following main treatments: liver transplantation or resection [4], transcatheter arterial chemoembolization (TACE) [5], radiofrequency ablation (RFA) [6], transarterial radioembolization (TARE) [7], and targeted systemic chemotherapy [8]. Surgical treatment is normally considered standard potentially curative treatment for early-stage HCC [9]. However, this is not a comprehensive HCC strategy, as most of the patients diagnosed at advanced stage have a poor prognosis [2]. Despite the numerous efforts to treat of HCC, the five-year recurrence rate remains high (~60%) after surgical treatment [10]. Therefore, the need is urgent to develop more effective treatments that can improve the long-term survival rate of HCC patients. 

The pathogenesis of liver cancer is complicated [11]. Multiple risk factors, such as age and race [12], hepatitis C virus [13], and hepatitis B virus [14], are considered to contribute to the abnormal expression of genes associated with initiation or progression of HCC. One of the strategies under investigation for the treatment of HCC is to discover essential genes contributing to the initiation or progression of HCC. Recent evidence shows that a variety of genes are crucial regulators of HCC. For instance, elevation of chemokine (C-X-C motif) ligand 17 (CXCL17) expression was demonstrated to enhance HCC cell metastasis and suppress autophagy via the LKB1 (a serine/threonine kinase)-AMPK (a central metabolic sensor) pathway [15]. The analysis of tumor necrosis factor-α (TNF-α) expression in 62 HCC patients indicated that TNF-α may serve as a potential HCC therapeutic target to overcome sorafenib resistance [16]. Overexpression of hepatic cyclooxygenase-2 in mice was reported to induce spontaneous formation of HCC [17]. Overall, many genes have abnormal expressions in HCC. The HCC-associated genes, along with a subset of their neighbors, are generally developed into a gene interaction network. Some researchers are interested in studying HCC from the perspective of gene interaction networks rather than assess the genes individually. The availability of multiple network topological analytic methods has enabled the identification of a number of hub genes with important pathological significance [18,19,20]. 

HCC forms a highly heterogeneous tumor, and various alterations in gene expression can promote the initiation and progression of this cancer. Zhang et al. found the top 10 hub genes for HCC are *ALB*, *TGFB1*, *GMPS*, *ACACA*, *KRAS*, *EGFR*, *STAT3*, *ERBB2*, *BCL2*, and *CD8A* [11]. In contrast, another study showed that *JUN*, *EGR1*, *MYC*, and *CDKN1A* are hub genes of HCC [21]. Heterogenicity may contribute to the differences in these two studies. Most of the previous studies were performed with only one micro-array gene expression profile which may not fully detect all potentially HCC hub genes. Attempting to find more representative genes of HCC, we integrally analyzed three micro-array gene expression profiles to detect genes that may serve as therapeutic target genes of HCC.

## 2. Materials and Methods 

### 2.1. Data Selection

Three micro-array gene expression profiles (GSE87630, GSE84598, and GSE89377) were selected from the Gene Expression Omnibus (GEO) repository [11]. GSE87630 is based on the GPL6947 platform (Illumina HumanHT-12 V3.0 expression beadchip, San Diego, CA, USA) and GSE84598 is based on platform GPL10558 (Illumina HumanHT-12 V4.0 expression beadchip, San Diego, CA, USA). The platform information for GSE89377 is as follows: GPL6947, Illumina HumanHT-12 V3.0 expression beadchip (San Diego, CA, USA). A total of 196 samples were selected for differentially expressed analysis after a careful review of the sample data. As all of the data were downloaded from GEO, we did not perform any human or animal experiments.

### 2.2. Differential Expression Analysis of the Selected Gene Expression Profiles 

GEO2R (https://www.ncbi.nlm.nih.gov/geo/geo2r/) [11], an interactive online analysis tool, was used to detect the differentially expressed genes (DEGs) between HCC and normal liver samples. The differential expression was analyzed separately on each profile. All parameters were set to default. The genes that met the cutoff criteria (adjusted *p*-value < 0.05 and |log fold-change (logFC)| > 1) were designated as DEGs. An online analysis tool (https://bioinfogp.cnb.csic.es/tools/venny/?/index.html) was applied to draw the Venn diagrams of up- or down-regulated genes and to detect the intersecting part of the DEGs.

### 2.3. GO (Gene Ontology) and KEGG (Kyoto Encyclopedia Genes and Genomes) Enrichment Analysis of Common DEGs

GO and KEGG pathway enrichment analyses were conducted with the intersecting DEGs. GO analysis is a regular bioinformatics method used for identifying representative biological attributes of large‑scale transcriptomic or genomic data. It provides valuable information about molecular function (MF), biological process (BP), and cellular component (CC). KEGG is a database that links DEGs with manually drawn reference pathways. The GO and KEGG pathway enrichment analyses of DEGs were conducted using the Database for Annotation, Visualization and Integrated Discovery, (DAVID, https://david.ncifcrf.gov/) [22]. *p* ˂ 0.05 and gene count ˃ 10 were the cutoff criteria for the GO and KEGG functional enrichment analysis.

### 2.4. PPI Network Construction and Identification of Hub Genes

The functional interactions between common DEGs encoded proteins may provide valuable information on molecular activities important for carcinogenesis. In this study, a protein-protein interaction (PPI) network of the common DEGs encoded proteins was constructed via the STRING database (http://string-db.org) [23]. The cut-off criterion for PPI construction was set to interaction score of ≥0.4. The PPI network interaction data were then visualized with Cytoscape software (version 3.4.0, http://cytoscape.org) [11]. As a Java plugin app for Cytoscape, CytoHubba provides a user-friendly interface facilitating the analysis of complex networks with topology methods. A total of 11 topological algorithms were available for the identification of hub genes in a complex PPI network. The Maximal Clique Centrality (MCC) algorithm performs better performance in predicting hub genes in PPI networks compared with the rest of the topological algorithms. Thus, we selected the MCC algorithm to identify HCC hub genes. 

### 2.5. Validation of HCC Hub Gene Expression via Oncomine Database

Oncomine is a web-based data-mining platform as well as a cancer-profiling database that aims to facilitate discovery from genome-wide expression analyses (https://www.oncomine.org) [24]. The database includes 715 gene expression data sets and gene expression data from 86,733 normal and cancer tissues. It has widely been employed in differential expression analysis or co-expression analysis between cancer and normal tissues, spanning most of the cancer types and subtypes [24,25]. Here, we obtained the mRNA expression value of the HCC hub genes based on analysis of independent gene expression profiles data with Oncomine. The gene expression data were then visualized and statistically analyzed with GraphPad software (version 5, GraphPad Software, Inc., San Diego, CA, USA).

### 2.6. Survival Analysis 

The Kaplan–Meier plotter (KMplot, http://www.kmplot.com/analysis) is a database that can be used to test the role of 54,675 genes on the survival time of patients with 10,293 cancer specimens [26]. The specimens included in this database originated from 1648 ovarian, 5143 breast, 1065 gastric, 2437 lung, and 364 HCC cancer patients. The Kaplan–Meier plotter database is generally recognized as an effective approach for gene prognosis assessment in many types of cancer. In addition, the HCC genes associated survival analysis in this database is based on a paper published by Menyhárt et al. (2018) [27]. Therefore, the Kaplan–Meier plotter database is a reliable approach for the survival analysis of HCC hub genes. Here, the liver tissue mRNA RNA-seq data from the Kaplan–Meier plotter were used to assess the prognostic role of hub genes in HCC patients. For each hub gene, the specimens were divided into two groups based on the median values of mRNA expression and *p* ˂ 0.05 was considered a statistically significant result between these two groups.

## 3. Results

### 3.1. Identification of DEGs with GEO2R

The DEGs were identified by comparing HCC samples with normal liver samples. A total of 2225 DEGs were identified from GSE84598 based on the cutoff criteria of *p* < 0.05 and |logFC| > 1, including 1017 up-regulated genes and 1208 down-regulated genes. For GSE87630, 1164 DEGs were screened out, including 395 up-regulated genes and 769 down-regulated genes. The GSE89377 differential expression analysis resulted in 578 DEGs with 148 up-regulated and 430 down-regulated genes. To identify the intersection among the three DEGs profiles, a Venn diagram was plotted followed the differential expression analysis, which resulted in 51 common up-regulated genes (Figure 1a) and 201 common down-regulated genes (Figure 1b) among GSE87630, GSE84598, and GSE89377.

### 3.2. GO and KEGG Enrichment Analysis 

DAVID online enrichment analysis was conducted. *n* > 10 and *p* < 0.5 were set as the cutoff criteria for significant enrichment. For GO BP, the common DEGs were significantly enriched in oxidation-reduction process, inflammatory response, proteolysis, immune response, cell division, complement activation, mitotic nuclear division, xenobiotic metabolic process, cell surface receptor signaling pathway, and response to drug (Figure 2a). For GO CC, the common DEGs were significantly enriched in extracellular region, organelle membrane, extracellular space, extracellular exosome, blood microparticle, perinuclear region of cytoplasm mitochondrial matrix, integral component of plasma membrane, endoplasmic reticulum membrane, and cytosol (Figure 2b). For GO MF, the common DEGs were significantly enriched in oxidoreductase activity, monooxygenase activity, heme binding, iron ion binding, serine-type endopeptidase activity, enzyme binding, receptor binding, calcium ion binding, and protein homodimerization activity (Figure 2c). As shown in Figure 2d, the most significantly enriched KEGG pathways included cell cycle, metabolic pathways, chemical carcinogenesis, and complement, and coagulation cascades.

### 3.3. PPI Network Construction and Identification of Hub Genes Based on Network Topological Analysis

The 252 common DEGs of the results from the differential expression analysis of the HCC gene expression profiles were used to construct a PPI network with the STRING database. The constructed PPI network includes a total of 250 nodes and 887 edges. CytoHubba, an app for Cytoscape software, is generally used to predict hub nodes from a given network based on 11 topological algorithms. Here, the Maximal Clique Centrality (MCC) algorithm of CytoHubba was employed to identify the hub genes. A total of 10 hub genes were finally identified: *CDCA5*, *OIP5*, *TOP2A*, *PRC1*, *ASPM*, *NUSAP1*, *UBE2C*, *CDC20*, *CCNA2*, and *KIF20A* (Figure 3). LogFC is the log difference of actual expression values results from expression analysis of two groups of tissue samples. The logFC of the HCC hub genes is displayed in Table 1, revealing the expression alterations of hub genes in HCC samples compared with normal liver tissue. A positive logFC indicates up-regulation of the hub genes, whereas a negative logFC suggests down-regulation of genes. All the logFC values of the HCC hub genes were positive, indicating an overexpression of the hub genes in HCC tissues.

### 3.4. Validation of Hub Genes Expression Level with Data Mining Using Oncomine Database

Based on data mining of the GSE6764 dataset from the Oncomine database, the gene expression values of *ASPM*, *CCNA2*, *CDC20*, *CDCA5*, *KIF20A*, and *OIP5* were obtained. Similarly, the gene expression values of *NUSAP1*, *PRC1*, *TOP2A*, and *UBE2C* were also obtained via data mining of the GSE14520 dataset from the Oncomine database. The gene expression values of the hub genes were then statistically analyzed and visualized with GraphPad software 5.0 (San Diego, CA, USA). Our results indicated that the HCC tissues have significantly higher levels of mRNA expression in terms of hub genes compared to normal liver tissues (Student’s *t*-test, *p* < 0.05; Figure 4a–j).

### 3.5. Survival Analysis

The Kaplan–Meier plotter was used to evaluate the role of the identified hub genes on the prognosis of HCC. A total of 364 liver samples were available for current survival analysis. We found that overexpression of these hub genes is associated with a significant reduction in survival time of HCC patients (*p* < 0.05; Figure 5a–j).

## 4. Discussion

HCC, a major health threat to people worldwide, is one of the leading causes of cancer mortality [28]. It occurs more frequently among men than women, with increasing incidence rates in almost all countries, particularly in East Asia [29]. The etiologies of HCC include metabolic diseases [30], alcohol-related cirrhosis [31], hepatitis B virus (HBV) [32], and hepatitis C virus (HCV) [33]. HCV-associated hepatitis develops into HCC via a myc-unregulated mechanism [34], whereas non-alcoholic steatohepatitis-dependent disease is not involved in the expression of myc [35]. The large difference in etiologies might contribute to the heterogeneity of HCC. 

So far, no targeted therapies have been proven to be effective against HCC. Due to the heterogeneity of HCC, the current targeted cancer drugs only selectively kill some tumor cells, leaving a small part of the surviving tumor cells almost unaffected, which gradually develop into the main body of tumor [36]. Therefore, HCC studies should cover as many cases and genes as possible and this increase in sample size or gene numbers may allow us to detect a large number of genes highly associated with HCC. 

The development of microarray technology has allowed to measure numerous gene expression aberrations simultaneously. Currently, many gene expression profiles are freely available in GEO. In this study, three gene expression profiles were selected from GEO for differential expression analysis. Venn analysis resulted in a total of 51 common up-regulated genes and 201 down-regulated genes. GO enrichment analyses indicated that the common DEGs are closely linked to a series of cancer functional events. The function enriched KEGG pathways are associated with cell cycle. The top 10 hub genes include *CDCA5*, *OIP5*, *TOP2A*, *PRC1*, *ASPM*, *NUSAP1*, *UBE2C*, *CDC20*, *CCNA2*, and *KIF20A*. Cai et al. showed that *CDC20*, *AURKB*, *BIRC5*, *RRM2*, *MCM2*, *PTTG1*, *CDKN2A*, *NEK2*, *CENPF*, *RACGAP1*, *GNA14*, and *CDCA5* are hub genes that may be important for diagnosis, clinical intervention, and prognosis of HCC [37]. *TOP2A*, *NDC80*, *FOXM1*, *HMMR*, *KNTC1*, *PTTG1*, *FEN1*, *RFC4*, *SMC4*, and *PRC1* were reported as being the top 10 core HCC genes in another study, sharing two genes with our study [38]. Heterogeneity, sample size, and methods of identifying hub genes may contribute to the differences in hub genes. Six of the hub genes (*OIP5*, *ASPM*, *NUSAP1*, *UBE2C*, *CCNA2*, and *KIF20A*) were identified for the first time in comparison with previous reports.

In the current study, we combined three gene expression datasets to find the hub genes important for HCC progression or prognosis. Three gene expression profiles from different regions were used and the sample size was larger compared to some of the previous studies, which might have helped to minimize the adverse effects caused by heterogeneity and small sample size. Cell-division cycle-associated 5 (*CDCA5*) is believed to play an essential role in the accurate separation of sister chromatids throughout the S and G2/M phases of the cell cycle [39,40]. A clinical study confirmed *CDCA5* overexpression is associated with poor prognosis in patients with HCC [41]. Opa interacting protein 5 (*OIP5*) is reported to regulate growth and metastasis of HCC via the Protein Kinase B (AKT)/mammalian target of the rapamycin (mTOR) signaling pathways [42]. High expression of DNA topoisomerase II alpha (*TOP2A*) was detected in 72.5% of HCC tumor tissues [43] and an increase in TOP2A expression is correlated with shorter survival of patients and chemoresistance [44]. Protein regulator of cytokinesis 1(*PRC1*) overexpression in HCC cells can cause enhanced chemoresistance and attenuation of apoptosis for patients who received chemotherapy [45]. Abnormal spindle-like microcephaly-associated protein (*ASPM*) overexpression has been shown to be a molecular marker predicting enhanced metastatic potential of HCC with poor prognosis [46]. Of the HCC patients, 54.1% were observed to have high expression of nucleolar- and spindle-associated protein 1 (*NUSAP1*) and this overexpression of NUSAP1 is associated with a significantly lower survival rate of patients [47]. microRNA 193a-5p was proven to suppress mice hepatocarcinogenesis via the regulation of *NUSAP1* level [48]. So far, little information is available regarding the contribution of ubiquitin conjugating enzyme E2 C (*UBE2C*) to HCC development. Up-regulation of cell division cycle 20 (*CDC20*) in tumor tissues can be used to predict worse overall survival in HCC patients [49]. In vitro experiments showed cyclin A2 (*CCNA2*) contributes to the tumorigenesis of HCC [50]. In comparison with normal liver tissues, HCC tissues generally have high levels of kinesin family member 20A (KIF20A) expression with worse survival outcomes [51]. Overall, most of the hub genes identified in this study had a high level of expression with poor prognosis. 

To further validate the expression level of the hub genes, we performed Oncomine-database-based mRNA expression-level analysis of the hub genes which showed that all the hub genes have significantly higher levels of expression in comparison with normal liver tissues. The results suggest that the hub genes we screened out are reliable biomarkers of HCC. 

Survival analysis showed that high expression of these hub genes is associated with a significant reduction (*p* < 0.05) in overall survival time in HCC patients. Overexpression of these hub genes, especially the six hub genes with no previous reports, may serve as unfavorable prognostic factors for HCC patients.

## 5. Conclusions

In this study, 10 HCC hub genes were identified, with few reports of six of them. Overexpression of the hub genes is significantly associated with decreased survival time in HCC patients. The hub genes play a role in disease onset or progression, which might be novel therapeutic targets for the treatment of HCC for patients with similar genetic alterations. 

## Figures and Tables

**Figure 1 ijerph-17-01053-f001:**
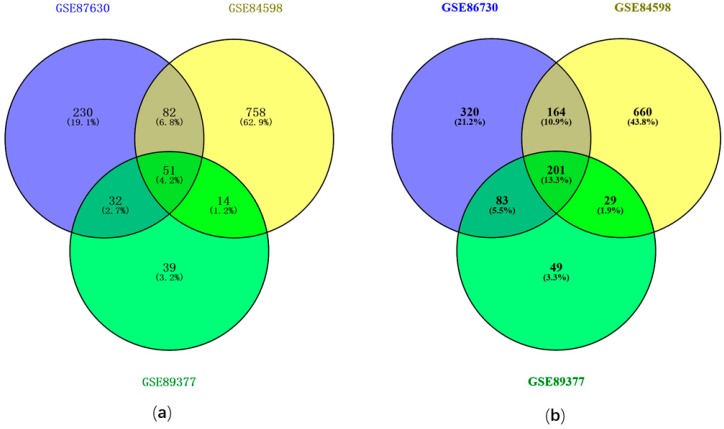
Venn diagram of common differentially expressed genes (DEGs) from integral analysis of three independent gene expression profiles: (**a**) up-regulated genes and (**b**) down-regulated genes from differential expression analysis of three gene expression profiles.

**Figure 2 ijerph-17-01053-f002:**
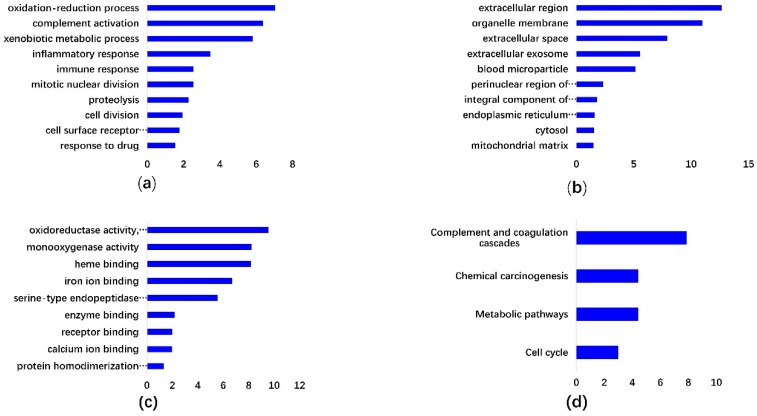
The significantly enriched Gene Ontology (GO) terms and Kyoto Encyclopedia Genes and Genomes (KEGG) pathways of hepatocellular carcinoma (HCC) differentially expressed genes. (**a**) GO biological process, (**b**) GO cellular component, and (**c**) GO molecular function enrichment analysis results of DEGs. (**d**) GO KEGG pathway enrichment analysis of DEGs in HCC.

**Figure 3 ijerph-17-01053-f003:**
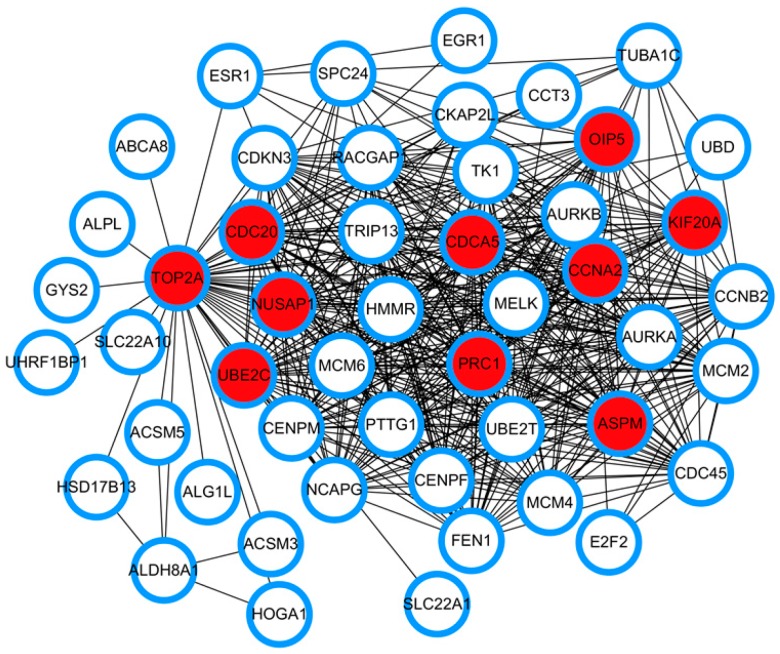
The protein-protein interaction network of the hub genes and the DEGs that directly interact with them. Each node represents a gene and each edge represents a direct interaction between two genes. The nodes filled with red in the center represent hub genes; the nodes filled with white represent the genes that interact directly with hub genes.

**Figure 4 ijerph-17-01053-f004:**
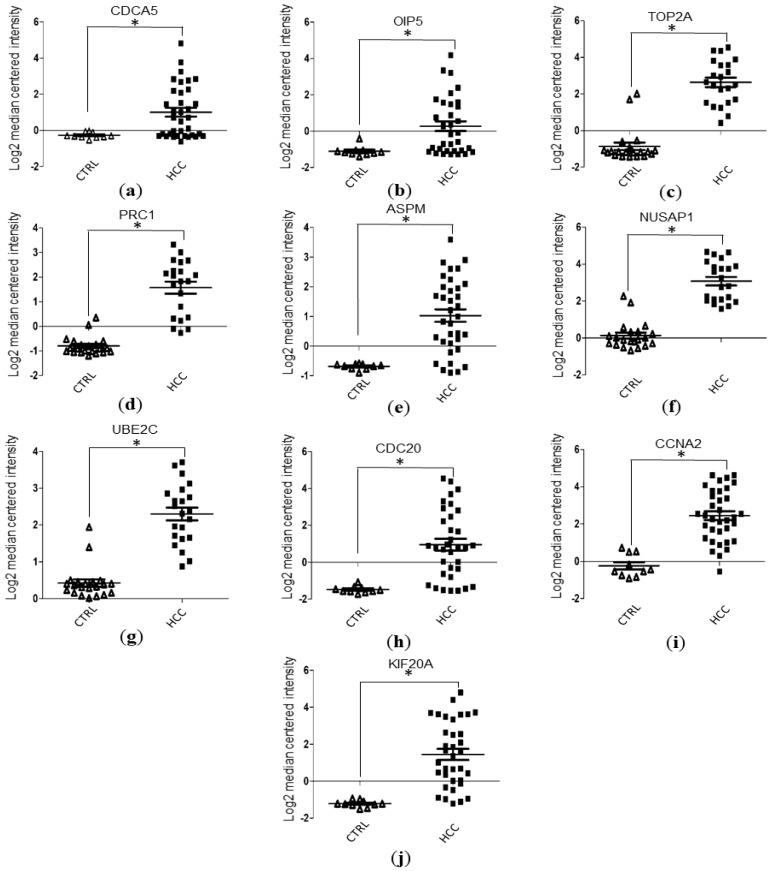
Validation of HCC hub gene expression levels with data mining of the Oncomine database: (**a–****j**) The gene expression level of the hub genes between HCC tissues and normal liver tissues. *, a significant statistical difference was observed between HCC and normal liver tissues. Based on the analysis of the GSE6764 dataset, *ASPM*, *CCNA2*, *CDC20*, *CDCA5*, *KIF20A*, and *OIP5* have significantly higher levels of expression in HCC samples compared to normal liver tissues (Student’s *t*-test, *p* < 0.05). Similarly, *NUSAP1*, *PRC1*, *TOP2A*, and *UBE2C* have significantly higher levels of expression in HCC samples compared to normal liver tissues (Student’s *t*-test, *p* < 0.05) via analysis of GSE14520 dataset with the Oncomine database (Student’s *t*-test, *p* < 0.05).

**Figure 5 ijerph-17-01053-f005:**
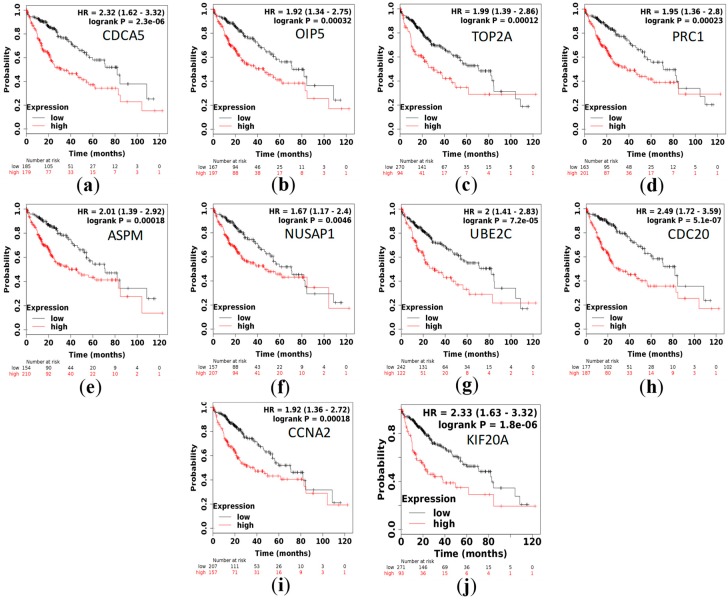
Kaplan–Meier survival analyses of the HCC hub genes. (**a****–j**) Overexpression of these HCC hub genes is associated with a significant reduction in overall survival time in HCC patients.

**Table 1 ijerph-17-01053-t001:** Log fold-change (logFC) of hub gene expression values in HCC samples compared with normal samples.

Hub Genes	GSE87630	GSE89377	GSE84598
*CDCA5*	2.28	1.9	2.55
*OIP5*	1.32	1.29	1.72
*TOP2A*	3.15	2.33	3.21
*PRC1*	2.74	1.81	2.56
*ASPM*	1.53	1.61	2.25
*NUSAP1*	1.82	1.67	1.53
*UBE2C*	1.88	2.4	2.77
*CDC20*	3.01	2.33	3.09
*CCNA2*	1.56	1.03	1.82
*KIF20A*	2.22	1.37	3.76

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
