# Peer review of "Identification of Potentially Therapeutic Target Genes of Hepatocellular Carcinoma"

_ijerph, 2020, doi:10.3390/ijerph17031053_

Round 1
Reviewer 1 Report
In this paper, analysis of gene expression profiles of HCC was performed to obtain a list of common differentially expressed genes (DEGs). Bioinformatic analyses of DEGs allowed hub gene identification that was followed by protein expression validation and survival analysis. A total of 51 common up-regulated DEGs and 201 down-regulated DEGs were identified after deferentially expressed gene analysis. GO and KEGG enrichment analyses indicated that these common DEGs were linked to a series of cancer events. Ten hub genes were finally identified and six of them (OIP5,ASPM,NUSAP1,UBE2C,CCNA2,KIF20A) were novel hub genes of HCC. Survival analysis indicated that hub genes overexpression was associated with a significant reduction of survival time in HCC patients.
This is an interesting study, well performed the results of which will be useful for further research on HCC therapy. I have no major objection. Larger characters in the scriptures in Figures 2 and 5 could facilitate reading.
Author Response
Thank you for your valuable and helpful comments. We have studied the comments carefully and revised our paper according to the comments.
Larger characters in the scriptures in Figures 2 and 5 could facilitate reading.
Answer: We agree with the comments. To facilitate reading, characters in figure 2 are now present in bold and larger characters. Text in figure 5 are also displayed in larger and bold characters.
Thank you once again for your comments.
Reviewer 2 Report
In their paper entitled “Identification of potentially therapeutic target genes of hepatocellular carcinoma” the Authors examined, by three different gene expression profile generators, gene expression in hepatocellular carcinoma (HCC), looking for hub genes. They report identification of six novel hub genes of HCC which might represent, in the future, new therapeutic targets for HCC therapy.
The paper is of interest, but some points should be addressed before acceptance:
A general revision of English is recommended in order to correct spelling errors and mistakes in verb use and/or in sentence construction. See, for example:1a. Abstract; lines 12-13: ‘gene expression profiles were subject to…’, and ‘the DEGs were then undergone’;
1b. Discussion, line 233: ‘As the sample size is bigger and from different regions.’: no main sentence present;
Please, write the meaning of acronyms the first time they are cited, including the Abstract; In paragraph 2.6 (Methods), the Authors write that they used the Kaplan–Meier plotter in order to evaluate the role of the identified hub genes in the prognosis of HCC, and the patient survival time. As they write, ‘the specimens included in this database were originated from 1,648 ovarian, 5,143 breast, 1,065 gastric and 2,437 lung cancer patients, with an average follow-up of 40 / 69 / 33 / 49 months respectively’. Thus, no HCC specimen seems to have been included in this database. Later on, in the ‘Results’ Section, paragraph 3.5, the Authors write: ‘A total of 364 liver samples were available for the purpose of current survival analysis. This is a bit confusing for the Reader, and should be discussed to justify the use of this database in the search for any relationship between HCC hub genes and survival; At least an attempt to evaluate the actual expression of these genes in HCC samples should add practical significance to these data.Author Response
Thank you for your kind and valuable comments. We have studied the comments carefully and revised our paper according to the comments.
A general revision of English is recommended in order to correct spelling errors and mistakes in verb use and/or in sentence construction. See, for example: 1a. Abstract; lines 12-13: ‘gene expression profiles were subject to…’, and ‘the DEGs were then undergone’. Discussion, line 233: ‘As the sample size is bigger and from different regions.’: no main sentence present;
Answer: In this version of the manuscript, a general revision of English is performed to correct the spelling errors, mistakes in verb use and/or in sentence construction through English language editing service provided by MDPI (English editing ID: english-16126).
“Abstract;lines 12-13: ‘gene expression profiles were subject to…’, and ‘the DEGs were then undergone’.” has been corrected as “The gene expression profiles were analyzed with GEO2R, an interactive web tool for gene differential expression analysis, to identify common differentially expressed genes (DEGs). Functional enrichment analyses were then conducted followed by a protein–protein interaction (PPI) network construction with the common DEGs. The PPI network was employed to identify hub genes, and the expression level of the hub genes was validated via data mining the Oncomine database. Survival analysis was carried out to assess the prognosis of hub genes in HCC patients.”
“Discussion, line 233: ‘As the sample size is bigger and from different regions.’: no main sentence present;” has been modified as follows “ has been modified as “Three gene expression profiles from different regions were used and the sample size was bigger compared to some of the previous studies, which might have helped to minimize the adverse effects caused by heterogeneity and small sample size.”
Please, write the meaning of acronyms the first time they are cited, including the Abstract.
Answer: We have checked each section of the manuscript and added the meaning of acronyms the first time they are cited, including the Abstract.
In paragraph 2.6 (Methods), the Authors write that they used the Kaplan–Meier plotter in order to evaluate the role of the identified hub genes in the prognosis of HCC, and the patient survival time. As they write, ‘the specimens included in this database were originated from 1,648 ovarian, 5,143 breast, 1,065 gastric and 2,437 lung cancer patients, with an average follow-up of 40 / 69 / 33 / 49 months respectively’. Thus, no HCC specimen seems to have been included in this database. Later on, in the ‘Results’ Section, paragraph 3.5, the Authors write: ‘A total of 364 liver samples were available for the purpose of current survival analysis. This is a bit confusing for the Reader, and should be discussed to justify the use of this database in the search for any relationship between HCC hub genes and survival;
Answer: We agree with the comments. As we did not give a description of HCC specimen in the last paragraph of methods section, this may cause a bit confusing for the readers. “the specimens included in this database were originated from 1,648 ovarian, 5,143 breast, 1,065 gastric and 2,437 lung cancer patients, with an average follow-up of 40 / 69 / 33 / 49 months respectively” has now been corrected as “The specimens included in this database originated from 1648 ovarian, 5143 breast, 1065 gastric, 2437 lung, and 364 HCC cancer patients.”
To justify the use of Kaplan Meier plotter database in the search for possible relationship between HCC hub genes and survival, the following sentence has been added to the last paragraph of methods section “The Kaplan–Meier plotter database is generally recognized as an effective approach for gene prognosis assessment in many types of cancer. In addition, the HCC genes associated survival analysis in this database is based on a paper published by Menyhárt et al (2018). Therefore, Kaplan Meier plotter database is a reliable approach for the survival analysis of HCC hub genes.”
At least an attempt to evaluate the actual expression of these genes in HCC samples should add practical significance to these data.
Answer: The following sentences together with a table reflecting the actual expression of these genes in HCC has been added in relevant parts of result section. “LogFC is the log difference of actual expression values results from expression analysis of two groups of tissue samples. The logFC of the HCC hub genes is displayed in Table 1, revealing the expression alterations of hub genes in HCC samples compared with normal liver tissue. A positive logFC indicates up-regulation of the hub genes, whereas a negative logFC suggests down-regulation of genes. All the logFC values of the HCC hub genes were positive, indicating an overexpression of the hub genes in HCC tissues.”
Reviewer 3 Report
This manuscript by Li and Xu addresses the question of whether specific genes are differentially expressed in patients with hepatocellular carcinoma. The authors used 3 publicly available GEO microarray datasets to identify genes that are overexpressed or downregulated in all three datasets. The authors then used GO and KEGG pathway analysis to identify potential cellular pathways that are disrupted by the misregulation of these genes. Finally, the authors used publicly available datasets to perform a Kaplan-Meier survival analysis and found that the overexpression of 10 genes led to decreased survival of patients with HCC compared to healthy controls.
Overall, this study only re-iterates what is already known about the misregulation of some genetic factors in HCC. Similar to what the authors state multiple times in this manuscript, the heterogeneity in HCC gene expression profile can explain differences between gene expression datasets obtained from HCC patients. Therefore, it is quite possible that there is no 'common' gene expression profile in HCC and several genes can play in a role in disease onset or progression based on the patient's genetic and environmental factors. My suggestion to the authors is to tone down some of their conclusions and present a more balanced view of their analysis.
Author Response
Thank you for your helpful and valuable comments. We have studied the comments carefully and response throughout the paper according to the comments.
Overall, this study only re-iterates what is already known about the misregulation of some genetic factors in HCC. Similar to what the authors state multiple times in this manuscript, the heterogeneity in HCC gene expression profile can explain differences between gene expression datasets obtained from HCC patients. Therefore, it is quite possible that there is no 'common' gene expression profile in HCC and several genes can play in a role in disease onset or progression based on the patient's genetic and environmental factors. My suggestion to the authors is to tone down some of their conclusions and present a more balanced view of their analysis.
Answer: We agree with the comments. We have toned down some of the conclusions present in abstract as well as other sections of the manuscript.
Abstract; lines 20-21,“Ten hub genes were finally identified and six of them (OIP5,ASPM,NUSAP1,UBE2C,CCNA2,KIF20A) were reported for first time to be novel hub genes of HCC. ” has been corrected as “We finally identified 10 hub genes, 6 of which (OIP5, ASPM, NUSAP1, UBE2C, CCNA2, and KIF20A) are reported as novel HCC hub genes. ”
Abstract; lines 25-26,“This study identified six novel hub genes of HCC which may be new potentially therapeutic targets for the development of HCC drugs.” has now been replaced with “We identified six novel HCC hub genes that might be therapeutic targets for the development of drugs for some HCC patients”.
The last sentence in paragraph 3 discussion section has been corrected as follows “Six of the hub genes (OIP5, ASPM, NUSAP1, UBE2C, CCNA2, and KIF20A) were firstly identified in comparison with that of previous reports.”
Discussion; lines 241-243. The second sentence in paragraph 4 has been replaced with “Three gene expression profiles from different regions were used and the sample size was bigger compared to some of the previous studies, which might have helped to minimize the adverse effects caused by heterogeneity and small sample size.”
Discussion; lines 241-243. The last sentence in the last paragraph of discussion has been corrected as “Overexpression of these hub genes, especially the six hub genes with no previous reports, may serve as unfavorable prognostic factors for HCC patients.”
Conclusions; lines 273-277.This section of the manuscript has been modified as follows “In this study, 10 HCC hub genes were identified, with few reports of six of them. Overexpression of the hub genes is significantly associated with decreased survival time in HCC patients. The hub genes play a role in disease onset or progression, which might be novel therapeutic targets for the treatment of HCC for patients with similar genetic alterations.”
Thank you once again for your comments.